# Assessing a Video-Based Intervention to Promote Parent Communication Strategies with a Deaf Infant: A Feasibility and Acceptability Study

**DOI:** 10.3390/jcm11185272

**Published:** 2022-09-07

**Authors:** Ciara Kelly, Ellen Crawford, Gary Morgan, Danielle Matthews

**Affiliations:** 1Division of Human Communication, Development and Hearing, University of Manchester, Manchester M13 9PL, UK; 2The Oxford Centre for Psychological Health, University of Oxford, Oxford OX3 7JX, UK; 3Department of Language and Communication Science, City, University of London, London EC1V 0HB, UK; 4Department of Psychology, University of Sheffield, Sheffield S1 2LT, UK

**Keywords:** intervention, deaf, hard of hearing, hearing loss, infant, infant–parent interaction

## Abstract

Infant–parent interaction forms the foundation for language learning. For the majority of deaf infants, hearing loss can impact access to, and the quality of communicative interactions, placing language development at risk. Support for families to meet the challenges faced during interaction is highly variable in the United Kingdom. In a step towards more standardized but tailorable family support, we co-produced an instructional, video-based intervention, testing for feasibility in terms of behavior change in seven communicative strategies and acceptability with 9 parents, forming study 1. Parents increased their use of the majority of behaviors and found content and delivery acceptable. However, further development was required to: (a) support use of semantically contingent talk and attention getting strategies to elicit infant attention, and (b) ensure the information was provided in a bite-size format that could be tailored to individual families. In study 2, the intervention was refined based on findings from study 1 and assessed for acceptability with 9 parents and 17 professionals, who reported similar high acceptability scores. Final refinements and modifications could be addressed in future interventions. The current studies provide a positive early step towards a standardized intervention to support communication that could be used in routine practice.

## 1. Introduction

Parent–child interaction in the early years plays a central role in a child’s language development, as the reciprocal social exchanges and synchrony between infant and parent form the context for language learning [1,2]. Indeed, the quality of infant–parent interaction predicts later language abilities [3,4,5], which has implications for later-life outcomes [6,7,8,9]. One group where interaction and subsequent developmental outcomes are at risk of disruption are deaf infants [10,11] (term used here to describe infants with any degree of hearing loss). Of the one in 1000 babies born with permanent hearing loss, approximately 95% are born to hearing parents [12], who, due to the mismatched hearing status, can struggle to make the intuitive behavioral adaptations necessary to scaffold communicative development [13]. For example, deaf infant–hearing parent dyads tend to demonstrate less flexibility, sensitivity, positive affect, and responsiveness to infant communicative and exploratory acts [14,15,16,17], and engage in more controlling and directive interactions [15,18,19].

Although varied in their communication choices (e.g., sign-supported speech, aural-oral approach), the majority of hearing parents typically have spoken language as the goal for their child [10]. Optimizing infant access to sound is critical to this goal but visual attention to communication is also important [20,21]. However, as supporting visual attention to communication is less intuitive for hearing parents [22,23,24], this group tend to make fewer attempts to elicit their infant’s visual attention before communicating [25,26]. This could be particularly problematic when infants engage in joint attention (i.e., a state of mutual awareness of shared attention towards the same object or event that is an essential foundation for language acquisition) [27,28,29]. Indeed, deaf infant–hearing parent dyads spend less time in joint attention compared to hearing dyads [30]. Equipping parents with effective behavior modifications could be an effective route to support parents [31].

### 1.1. Current Intervention Practice

Interventions that train parents to modify behaviors and implement specific strategies to support interaction have been shown to be effective [32,33,34,35]. This approach is in line with the international consensus statement of best practice for early intervention for deaf children [36], which places emphasis on promoting parents’ skills so that they are the positive agents of change in their child’s language development. However, many practitioners do not follow one formal parent support intervention but instead ‘cherry-pick’ strategies based on individual needs and family choices [37]. Although this needs-led approach is an important aspect of family-centered support, a lack of universal provision can lead to inconsistences in the support children with similar needs receive [32]. A more standardized but tailorable composite intervention might be useful for practitioners if the format and content were acceptable, and delivery was feasible and effective. 

### 1.2. The Present Studies

The aim of the two present studies was to test an intervention designed to promote specific communication strategies with the potential for use in routine practice. To that end, we first assessed the feasibility of an early iteration of a novel, instructional, video-based intervention in terms of success in appropriately changing parent behaviors, as well as parent acceptability in relation to content and delivery (study 1). We then further developed the intervention based on findings from study 1 and assessed the refined video-based intervention for acceptability with both parents and professionals (study 2).

## 2. Study 1

### 2.1. Materials and Methods

#### 2.1.1. Study Design

The present research is a pre-post, single group intervention feasibility study. Data was collected before and after administration of an instructional video-based intervention designed to change multiple parent communicative behaviours (see measurements and coding section for details).

#### 2.1.2. Participants

Nine participants were recruited across England and Scotland through the UK’s National Deaf Children’s Society (NDCS) database of families who have a deaf child as part of two concurrent studies; the present study and another investigating infant social communicative skills [38]. Two parents were of the same child in the same household (participants 6a and 6b). Participant demographics are detailed in Table 1. The eligibility criteria were determined by the latter study requiring that infants were full-term (born no more than 3 weeks before due date), with a birth weight over 2.5 kg, and no other known disabilities or developmental delays. Participants provided informed consent and received an age-appropriate book for their child and GBP 30.

#### 2.1.3. Procedure

The study involved two home visits two days apart. During the first visit, demographic information was collected following consent. Infant–parent dyads were then video recorded in free play together for 25 min during which, parents were asked to play with their infants as they normally would. Recordings were taken from two different camera angles and without the researcher present. Following this baseline recording, the intervention was administered, which involved the parent and researcher watching a video together. The researcher reiterated video instructions to try using the intervention strategies in everyday situations (see ‘Intervention’ section for full details) and provided parents with: (i) an opportunity to ask questions; (ii) a leaflet summarising the video content; and (iii) an intervention diary. Two days after visit one, the researcher returned to repeat the video-recorded play session post-intervention (where parents were asked to try using the intervention strategies), and to collect a parent questionnaire to assess acceptability of intervention content and method of delivery. Ethical approval was granted by the Department of Psychology’s Ethics Sub-Committee at The University of Sheffield.

#### 2.1.4. Intervention

The current intervention included strategies informed by parent–child interaction research briefly summarized below.

1.Strategies for Scaffolding Infant Skill Development

The interactive model of language intervention, which emphasises naturalistic interaction strategies [39,40], is a core feature of established interventions including the Hanen Program for Parents of Children with Language Delays [41]. Elements of the Hanen Program are used by some specialists supporting families with a deaf child [37]. A central aspect of this model involves being responsive and following the child’s lead [42]. A second aspect involves modifications to the structural features of linguistic input such as shorter utterances, slower speech tempo, and increased repetition (i.e., characteristics of infant directed speech; IDS). Increased parental self-repetition may be a particularly effective modification for deaf children [43,44,45], and is therefore included as a communication strategy alongside parental responsiveness.

2.Strategies for Making Communication Accessible: Deaf Parents as Role Models

The intuitive behavior modifications of fluent signing deaf parents are well adapted to supporting deaf children’s visual attention to communication [46,47,48,49,50], which is reflected in findings that deaf dyads spend more time engaged in joint attention than deaf infant–hearing parent dyads [25,48,51], and parallel hearing dyads [52]. The communication strategies of fluent signing deaf parents could therefore be beneficial as intervention strategies for hearing parents of deaf infants [24]. Specifically, the present intervention included visual-tactile attention getting strategies such as waving in the child’s line of vision, moving into their line of vision, and using touch (e.g., tapping), to elicit infant visual attention before communicating and to alert the infant to forthcoming communication [22,48,53]. We also included the strategy of persistence (i.e., repeated attempts) when eliciting infant attention as it leads to increased success but is used less frequently by hearing parents of deaf infants [54]. Finally, we included “accommodative strategies” (i.e., signs adapted to accommodate infant visual attention) for parents adopting communication approaches that include the use of sign (e.g., sign-supported speech). Accommodative strategies are typically used by fluent signing deaf parents to optimize visual accessibility of signs and provide the infant with the opportunity to make associations between referent and symbol with ease [22,55]. Strategies include displacing signs into the infant’s line of sight, or on or near the object the infant is attending to [22,46,56]. 

3.Additional Potentially Beneficial Strategies

Being at eye level when communicating with a deaf infant is considered to be beneficial for visual access to communication, including for spoken language (i.e., for speech-reading and non-verbal paralinguistic cues that support perception and understanding; [31,57,58]), and is therefore included as an intervention strategy.

4.Intervention Structure and Mode of Delivery

“Communicating with Your Baby” is a 21 min video intervention that describes various strategies to use during interaction with a deaf infant or toddler to support access to communication, scaffold developing communicative skills, and facilitate interaction. Strategies were described through the use of narration, illustrative images, and video clips. The script for the video was created using accessible language and reviewed by four experienced specialist Speech and Language Therapists. More detail on the specific intervention strategies can be found in Appendix A. Broadly, strategies were presented in four sections: (i) ‘Getting Down to Eye Level’ (i.e., changing position to increase infant visual and auditory accessibility); (ii) ‘Watching What Your Baby is Focusing on’ (i.e., noticing child’s focus of attention and commenting on it [semantically contingency talk]), which included using repetition when communicating; (iii) ‘Attracting Your Baby’s Attention’ (i.e., eliciting child’s attention before communicating with them and being persistent); and (iv) ‘Responding to Your Baby’s Attempts to Communicate with You’ (i.e., looking out for child’s attempts to communicate and responding using semantically contingent talk). Additionally, parents using sign were encouraged to use accommodative strategies when signing (e.g., placing signs near the referent).

#### 2.1.5. Measurements and Coding

1.Video Recorded Play Sessions

We developed a coding manual based on existing literature, previous coding manuals, and the research team’s skills and experiences (for the full coding manual, see Appendix A). The coding manual stipulates protocols to code for seven intervention strategies (described below) including: (i) visual accessibility of language; (ii) auditory accessibility of language; (iii) use of semantically contingent talk; (iv) repetition in child directed speech; (v) use of attention getting strategies (i.e., eliciting infant attention); (vi) persistence when using attention getting strategies; and (vii) use of accommodative strategies when signing. Coding manuals were iteratively piloted and refined by the research team before formal coding began to ensure they were accurately capturing the target behaviours.

Fifteen minutes of continuous video recording from the beginning of the session was coded for each participant using ELAN software [59], by trained research assistants who were blind to visit (i.e., pre- or post-intervention). Videos were initially coded for child directed speech (i.e., all speech directed to the child) as the majority of measures are based on parent utterances. Utterances were defined as units of child directed speech “bounded by grammatical closure or a pause of more than 2 s or transition in speaker” [60], which were then transcribed orthographically, following the CHILDES CHAT conventions [61]. 

(1)Accessibility of Language

Accessibility of language consisted of two elements, visual accessibility in the form of how easy it was for the child to see their parent’s face (determined by whether the parent was at eye level with the child or not) and auditory accessibility (determined by proximity to the child). 

Visual Accessibility

To measure visual accessibility, utterances were categorised into one of four positions: (i) eye level and face very accessible; (ii) eye level and face fairly accessible; (iii) not eye level but face fairly accessible; and (iv) not eye level and face difficult to access.

Auditory Accessibility

To measure auditory accessibility, utterances were coded into one of four proximity categories: (i) very close; (ii) close; (iii) not close; and (iv) far. Categories were based on the recommendation that spoken language within one to two meters of the child is the optimal range for hearing technologies to access speech [62]. Although encouraging a proximity of one to two meters when communicating was not explicitly stated in the intervention video (rather, to get down to eye level where it is easier for the child to see and hear the parent), we measured proximity in this context to explore whether parents naturally change their proximity when advised to ‘get down to eye level’.

(2)Semantically Contingent Talk

To measure parental responsiveness to infant focus of attention (i.e., the strategy ‘watching what your baby is focussing on’), all caregiver utterances were coded for semantic contingency on the infant’s focus of attention in the 5-s window preceding the utterance onset [63,64]. Following the coding scheme reported in McGillion et al. [65], utterances were coded as *contingent* if they referred to an object the infant focused on or referenced, or if they were related to an activity the infant was engaged in. Utterances were coded as *non-contingent* if they referred to an object or activity that the child had not attended to within the 5-s time window.

The strategy ‘responding to your baby’s attempts to communicate with you’ was not explicitly coded due to statistical constraints (small sample size precluding controlling for individual differences in infant communicative acts). However, as this strategy involved asking parents to respond to infant communication with semantically contingent talk, the semantic contingency coding scheme provided an indirect measure of this strategy. 

(3)Repetition in Child Directed Speech

Using a coding scheme based on similar studies [66,67,68], all instances of parent utterance repetition were identified and categorised according to repetition type with one of the following codes: (i) exact repetition (the utterance was repeated verbatim); (ii) exact + expansion (the utterance was repeated verbatim with additional information); (iii) partial repetition (one or more major units within an utterance were repeated); (iv) partial + expansion (one or more major units within an utterance were repeated with additional information); and (v) reframing (the repeated utterance reframed or paraphrased the source utterance, i.e., repetition of semantic content). Repetitions had to occur within 3 utterances following the source utterance and within a time window of 5 s to take into account the limitations of infant short-term memory and information processing capacities [66,68,69]. For the purposes of the present research, we collapsed each repetition type to form one composite total repeated utterances variable.

(4)Attention Getting Strategies

The authors developed a coding scheme to measure parent attention getting strategies, which was partially based on similar studies [22,54].

Attention Getting Episodes and Use of Individual Attention Getting Strategies

All instances where the parent attempted to elicit their child’s attention by engaging in an ‘attention getting episode’ were coded. An attention getting episode was defined as an attempt to elicit infant attention using attention getting strategies such as touch (which could involve a series of strategies or the use of one or two). Individual attention getting strategies within each episode were coded to provide a measure of total number of individual attention getting strategies used during the session and to compute persistence of attention getting episodes (detailed below). 

Function of Attention Getting Episodes

To explore whether parents used attention getting episodes as the intervention intended (i.e., to eliciting infant attention to self when communicating), episodes were categorised according to their function with one of the following codes: (i) eliciting attention to self; (ii) directing attention to object focus of attention (i.e., an object within the child’s focus of attention, following the semantic contingency coding scheme guidelines); (iii) directing attention to object non-focus of attention (i.e., an object that was not within the child’s focus of attention, following the semantic contingency coding scheme guidelines). 

Outcome of Attention Getting Episodes to Elicit Attention to Self

Episodes where the caregiver’s goal was to elicit infant attention to themselves were further coded for outcome (i.e., whether the infant looked or not). 

Persistence

Finally, parent persistence in eliciting their infant’s attention was scored as the average number of attention getting strategies per attention getting episode.

(5)Accommodative Strategies When Signing

All instances of sign-supported English (SSE) were coded for the parents who reported that they used this approach, which were then coded for occurrence of accommodative strategies. There were 3 accommodative strategies to code for: (i) sign displacement (e.g., signing next to referent); (ii) guiding child to make the sign (e.g., manipulating the child’s hands into a sign); and (iii) sign in child’s line of vision.

2.Reliability of Coding

To test the inter-rater reliability of the coding manual used to code intervention strategies, a sample of 10% of all recordings were second coded by an independent rater and agreement was calculated using Cohen’s Kappa and interpreted using guidelines from Viera and Garrett [70]. Agreement between raters for accessibility of language was excellent for visual, (κ = 0.97, *p* < 0.001), and auditory accessibility, (κ = 0.93, *p* < 0.001). There was also excellent agreement for coding semantically contingent talk, (κ = 0.95, *p* < 0.001). Rater agreement for repetition in child directed speech was also substantial, (κ ranging from 0.87 to 0.97, all *p*’s < 0.001). Agreement between raters suggests measures are reliable.

#### 2.1.6. Participant Reported Surveys

1.Demographic Survey

Demographic data was collected at baseline and included age of the child (at visit one), level of hearing loss, age when hearing aids were received, home communication approach(es), parent qualifications, primary caregiver, and Indices of Multiple Deprivation Decile (IMD; a measure of socioeconomic status). 

2.Intervention Acceptability Survey

A novel eleven item intervention acceptability survey comprising two sections (see Table 3 for items) was developed by the research team based on existing literature [71,72]. In the first section, participants indicated the extent to which they agreed or not with five items that were designed to measure parent perceptions of the specific strategies they were asked to use in the intervention. Participants indicated agreement using a 10-point Likert scale ranging from 0 (strongly disagree) through to 10 (strongly agree). For example, participants were asked to rate their endorsement of items such as, “*I enjoyed trying the tips from the video*”, and “*I did not find it difficult to use the tips in my daily routine*”. In the second section, participants were asked about their acceptability of the video itself. Participants were asked to respond to six items using a 10-point visual analogue scale. For example, one item asked whether “the video was… too short” (rated 0) through to “too long” (rated 10). Participants also answered open-ended questions at the end of each section that aimed to elicit acceptability feedback not covered by the survey items (e.g., “*Is there anything else you would like to tell us about your experience*”, and “*Do you think there are ways the video could be improved?*”). 

3.Participant Reported Intervention Diary

Participants were given an intervention diary (see Appendix A) to record thoughts and reflections in real-time when using the intervention strategies. Participants recorded the situations they were in when using the strategies (e.g., playtime, lunchtime, book reading, etc.), the duration they used the strategies for, and any other thoughts and reflections about the strategies.

#### 2.1.7. Approach to Analysis

A combination of descriptive and non-parametric statistics were used to investigate feasibility of parent behavior change from pre- to post-intervention, and, given the relatively small sample size, we report the median as a measure of central tendency as this is less vulnerable to outliers. To control for the total amount of speech, analyses are based on the proportions of total utterances that are coded within each level of the dependent variable. For example, for the variable *semantically contingent talk*, we report the proportion of total utterances that are *contingent* and *non-contingent* with pre- to post- changes in proportion as the unit of analyses. Additionally, we report the rank-biserial coefficient as a non-parametric measure of effect size alongside 95% confidence intervals. Although determining statistical significance was not the aim of the present feasibility study, confidence intervals that do not overlap zero were interpreted as statistically significant [73]. Finally, we report a process evaluation diagram to qualitatively describe individual participant-level experiences throughout the intervention.

### 2.2. Results

#### 2.2.1. Feasibility of the Intervention: Were Parents Able to Implement the Communication Strategies Successfully?

Table 2 presents an overview of the feasibility findings for all communicative behaviors, while Figure 1 presents a forest plot of the rank-biserial effect sizes changes from pre- to post-intervention for all behaviors. For participant-level raincloud plots showing changes from pre- to post-intervention for each behavior, see Appendix A.

1.Accessibility of Language(1)Visual Accessibility

Parents significantly increased the proportion of total utterances that were ‘eye level very accessible’ from pre- to post-intervention, (W = 0.00, Z = −2.67, effect size = −1.00, 95% CI = −1.00 to −1.00), whilst decreasing the proportion that were ‘not eye levelfairly accessible’, (W = 42.00, Z = 2.31, effect size = 0.87, 95% CI = 0.53 to 0.97), and also decreasing the proportion that were ‘not eye level difficult to access’, (W = 39.00, Z = 1.96, effect size = 0.73, 95% CI = 0.20 to 0.93). There was small, statistically non-significant increase from pre- to post-intervention in the proportion of utterances that were ‘eye level fairly accessible’, (W =19.00, Z = −0.42, effect size = −0.16, 95% CI = −0.71 to 0.52).

(2)Auditory Accessibility

Although no differences reached statistical significance, there was a medium-sized increase in the proportion of utterances that were ‘very close’, (W = 12.50, Z = −1.19, effect size = −0.44, 95% CI = −0.84 to 0.25), and a medium-sized decrease in the proportion that were ‘close’, (W = 33.00, Z = 1.24, effect size = 0.47, 95% CI = −0.23 to 0.85), from pre- to post-intervention. Finally, there were no changes in the proportion of utterances that were ‘not close’, (W = 14.00, Z = 0.00, effect size = 0.00, 95% CI = −0.68 to 0.68), and ‘far’, (W = 8.00, Z = −0.52, effect size = −0.24, 95% CI = −0.81 to 0.57), from pre- to post-intervention.

2.Semantically Contingent Talk

Parents demonstrated a small-sized (statistically non-significant) increase in the proportion of utterances that were ‘contingent’, (W = 17.00, Z = −0.65, effect size = −0.24, 95% CI = −0.76 to 0.45), and ‘non-contingent’, (W = 13.50, Z = −0.63, effect size = 0.25, 95% CI = −0.78 to 0.48), from pre- to post-intervention. 

3.Repetition in Child Directed Speech

Parents significantly increased the total number of repeated utterances from pre- to post-intervention, (W = 3.00, Z = −2.31, effect size = −0.87, 95% CI = −0.97 to −0.53).

4.Attention Getting Strategies

Parents significantly increased in frequency of attention getting episodes from pre- to post-intervention (W = 6.00, Z = −1.96, effect size = −0.73, 95% CI = −0.93 to −0.20), as well as in frequency of individual attention getting strategies (W = 4.00, Z = −2.19, effect size = −0.82, 95% CI = −0.96 to −0.40). To explore whether parents used attention getting episodes as the intervention intended (i.e., for the purpose of eliciting infant attention to self when communicating), the function of attention getting episodes was analyzed. There was a significant increase in the proportion of episodes used to elicit attention ‘to self’ (W = 4.00, Z = −2.19, effect size = −0.82, 95% CI = −0.96 to −0.40). There were no changes in the proportion of attention getting episodes used to redirect infant visual attention to an object that was in the infant’s focus of attention (i.e., ‘to object FOA’; W = 8.00, Z = 0.14, effect size = 0.07, 95% CI = −0.72 to 0.78) or to an object that was not in the infant’s focus of attention (i.e., ‘to object non-FOA’; W = 13.00, Z = −0.17, effect size = −0.07, 95% CI = −0.72 to 0.64).

The outcome of attention getting episodes that functioned to elicit infant attention ‘to self’ (i.e., whether these episodes were successful or not) was analyzed. There was a significant increase in the proportion of attention getting episodes to elicit infant attention to self that were successful (W = 5.50, Z = −1.75, effect size = −0.69, 95% CI = −0.93 to −0.08). Although the difference did not reach statistical significance, there was a small-sized increase in the proportion of attention getting episodes to elicit infant attention to self that were unsuccessful (W = 9.50, Z = −0.76, effect size = −0.32, 95% CI = −0.82 to 0.46). Finally, the average number of attention getting strategies per attention getting episode was analyzed as a score of persistence. Parents significantly increased in persistence when eliciting their infant’s attention from pre- to post-intervention (W = 0.00, Z = −2.37, effect size = −1.00, 95% CI = −1.00 to −1.00).

5.Accommodative Strategies When Signing

Although the intervention did *not* advise parents to increase their use of SSE, 4 out of the 5 parents who reported they used SSE did so. From pre- (median = 1) to post-intervention (median = 36) there was a large-sized increase that was not statistically significant (Z = −1.75, *p* = 0.125, r = −0.87, 95% CI = −0.98 to −0.34). We explored if parents adapted their signs to accommodate infant visual attention; however, only 2 parents used accommodative strategies. Participant 3, who was using both sign displacement and signing in the child’s line of vision pre-intervention, increased use post-intervention (frequencies from 4 to 15 and 1 to 6, respectively). Participant 4, who did not use any accommodative strategies pre-intervention, used both sign displacement and signing in the child’s line of vision post-intervention (post-intervention frequencies of 2 and 9, respectively). Neither parent used the strategy of guiding their child to make the sign.

#### 2.2.2. Parent Experiences Using Strategies Pre-, During, and Post-Intervention

To explore parent use of, experience using, and feelings towards the communicative strategies pre-, during, and post-intervention, we report a process evaluation flow diagram in Figure 2. During the intervention, all parents tried using the strategies in their daily routine. Parent diary entries include parents finding the strategies useful and enjoyable, with good responses from their child; however, parent 7 noted that carrying out the strategies is more time consuming. Additionally, parent 4 noted that it is difficult to elicit her child’s visual attention in all scenarios, including attention to signs, describing her child as “deliberately avoid[ing] eye contact”.

#### 2.2.3. Acceptability of the Intervention: Did Parents Find the Intervention Strategies and Method of Delivery Acceptable?

Findings on parent acceptability of the intervention video are presented in Table 3. Regarding acceptability of the intervention strategies, parents tended to report that they enjoyed using the strategies in the video (median = 8.5, IQR = 2.25), that they were not difficult to use in their daily routine (median = 9, IQR = 1.00), that they felt comfortable using the strategies (median = 10, IQR = 0.25), that the strategies helped them to communicate with their baby (median = 7.5, IQR = 2.25), and all participants stated that they would continue to use the strategies from the video going forwards (median = 10, IQR = 0.00). Parent responses to the open-ended question on additional thoughts are in line with these findings (see ‘Experience Using Video Strategies’ section in Figure 2). However, parent 4 reported finding signing difficult due to limited signs and reiterated her diary entry that she found it hard to elicit her child’s visual attention, noting that “he is almost avoiding my prompts”.

In terms of video delivery, parents reported that they felt the video length (median = 5.5, IQR = 2.50) and amount of information (median = 5, IQR = 0.75) was acceptable (although some participants reported that the intervention was too long, and contained too little information, see min-max scores in Table 3). Furthermore, parents reported that the video content was easy to understand (median = 10, IQR = 0.00) and looked professional (median = 8.5, IQR = 1.75). Finally, parents reported that using a video to deliver communication strategies was a good way to provide advice (median = 10, IQR = 1.50), and that they would recommend the video to other parents (median = 7.5, IQR = 2.50). Parent responses to the open-ended question on ways the video could be improved included a request for more tips, and comments that the illustrative clips were really useful, and that the repetition of strategies reinforced the point. Finally, parent 1 reported that some of the scenarios were not relatable to her (e.g., “one mum I think was deaf and they used sign”).

### 2.3. Discussion 

The present study assessed a novel, instructional, video-based intervention for feasibility in changing parent behaviors and acceptability in relation to content and delivery. The intervention was both feasible and acceptable; however, parent implementation of the strategy *semantically contingent talk* was less successful and eliciting infant visual attention before communicating was a challenge for parents. Additionally, implementing the strategies was considered to be more time consuming by one parent, and challenging for another, and the video itself was considered by some to be too long with too little information. 

#### 2.3.1. Intervention Feasibility

Parents increased their use of the majority of communication strategies in the intervention, particularly visual accessibility of language (i.e., getting down to eye level), use of repetition in child directed speech, use of attention getting strategies when eliciting their child’s visual attention, and persistence when using attention getting strategies. Not only did parents increase the proportion of their utterances at eye level, they were mostly in a position that was considered very accessible (i.e., minimal effort for the child to see their parent’s face). Furthermore, parents decreased the proportion of their utterances that were not at eye level. This pattern of decreasing less beneficial behaviors whilst increasing the target behavior was however, not the case for attention getting strategies. Although the intervention was successful at increasing parental use of attention getting strategies to elicit infant visual attention to themselves, parents did not decrease their use of these strategies to direct infant attention to an object that was not their child’s focus of attention. Further, at the participant level, three parents (6b, 7, and 8) actually increased their use of attention getting strategies to redirect infant attention. Given that hearing parents of deaf infants are known to be more directive of their child’s attention [15,18], and this type of interaction style is thought to have a negative impact on early language learning [74,75], further support may be required when it comes to implementing use of attention getting strategies.

When considering visual attention eliciting strategies, there may also be a need to take family heterogeneity into account, such as infant hearing loss levels and type of technology used, by striking a tailored balance between promoting visual and auditory accessibility to communication. As raised by parent 4, eliciting infant visual attention can be difficult when the child does not respond to prompts. Given that the use of attention getting strategies to elicit infant visual attention led to an increase in unsuccessful attempts as well as successful, it may be that the child had not yet learned his mother’s increased use of these strategies were attempts to direct his attention to visually perceive linguistic input [50]. However, Spencer and Harris report similar patterns when deaf parents elicit the visual attention of their deaf infants who have more experience with attention getting strategies [55]. An alternative possibility may be that the child does not need to rely as heavily on visually attending to communication given his moderate hearing loss, and use of hearing aids in conjunction with a frequency modulation (FM) system. Increasing the use of visual attention getting strategies may therefore be less beneficial for this particular family. Instead, there may be more of a need to focus on supporting auditory access. 

Although not an explicit strategy included in the intervention, we explored whether encouraging parents to get down to eye level would lead to a natural change in parent proximity to their child when communicating, particularly as this is important for visual and auditory access to communication [47,76]. There was a trend towards parents increasing the proportion of utterances produced very close to their infant (i.e., infant could touch parent’s face) and decreasing utterances produced close to the infant (approx. within 1 m). Parents may have decreased the latter utterance proximity in favor of the former, potentially due to getting down to eye level, given that changing their position so their head is aligned with their child’s could subsequently result in closer proximity (likely only when infants are not mobile). However, increasing eye level positioning may not guarantee a decrease in utterances produced at a distance considered *not close* (approx. 1–2 m) and *far* (approx. over 2 m), as there were no changes in either parent proximity. It is likely that utterances produced further away from the child were instances when the child was mobile. It may therefore be beneficial for future interventions to include being in closer proximity when communicating as a strategy but with care not to encourage parents to exert too much control and disrupt the natural interaction, which is a risk for deaf infant–hearing parent dyads [15,18].

One strategy that was less successful was parent implementation of semantically contingent talk, as there was a small (statistically non-significant) increase in the proportion of parent utterances that were contingent *and* non-contingent. It is possible that patterns may have been the same post-intervention as they were pre- but at an increased rate given that parents significantly increased repetition at the utterance level. Further, given that the proportion of non-contingent utterances were very low both pre-and post-intervention, with remaining utterances either contingent or other interaction regulating utterances (e.g., place filling expressions such as ‘good girl’), parents may have been at ceiling. Future interventions may want to consider including pre-assessment to determine whether it is necessary to advise the strategy of contingent talk (i.e., tailored intervention to meet individual needs). Indeed, this is an effective approach to intervention [77] and was a suggestion by parent 3. At the participant level, parents 4, 6b, and 8 demonstrated a slight increase in non-contingent utterances and a slight decrease in contingent utterances, which may be related to other intervention strategies. Specifically, the use of attention getting strategies, given that parents 6b and 8 increased use to redirect their child’s attention. Participant 4′s decrease could be an outcome of stress due to difficulties eliciting her infant’s attention, given the potential impact of stress on parental responsiveness [24]. These findings further emphasize the need for additional support when it comes to implementing the use of attention getting strategies.

Finally, the 5 parents who reported using SSE increased use post-intervention, even though the intervention did not advise this as a strategy. Parents may have increased use to attempt incorporating the intervention’s visually accommodative strategies; however, only 2 parents did so. It is unclear whether parents struggled to use these strategies or preferred not to, and whether the increase in use of SSE was due to attempts to use the accommodative strategies or because the intervention inadvertently encouraged use. Future interventions including sign-based strategies could explore this further. 

#### 2.3.2. Intervention Acceptability

Parent questionnaire responses suggest that the intervention content was acceptable as parents found trying the strategies enjoyable, not difficult, comfortable, and helpful when communicating with their child, and that they would continue to use the strategies. However, one participant’s responses suggest incorporating the strategies may not be as acceptable depending on the child’s response. For participant 4, the strategies were helpful and she would likely use them in future; however, she found them to be unenjoyable, difficult, and uncomfortable. The difficulties participant 4 faced when eliciting her child’s visual attention may underlie her negative responses, further emphasizing the need to consider family heterogeneity and find the right balance between encouraging visual and auditory access to language. An additional consideration may be to include regular “check-in” appointments to identify any potential adverse effects (e.g., finding the strategies time consuming or difficult) and provide further support. This may increase acceptability, limiting the risk of poor engagement further down the line [78].

In terms of method of delivery, the instructional video was considered to be acceptable as all parents felt the video was easy to understand, professional, and a good method to provide advice to parents. However, the method of delivery was less acceptable in terms of length and amount of information—the former being too long and the latter too little. One way to address this would be to provide a series of short, modular, instructional videos, each presenting a particular strategy or set of related strategies (such as attention getting strategies). By alternatively providing one large resource segmented into modules, parents can receive ample information that they do not need to cover in one lengthy session. This modular approach could additionally limit the time consuming nature of trying to implement all strategies at once, particularly when encouraged to try one strategy or set of related strategies at a time, until they become more habitual. 

## 3. Study 2

The aim of study 2 was to refine the intervention based on findings from study 1 and assess the refined intervention for acceptability with both parents and professionals. 

### 3.1. Tailoring Visual and Auditory Strategies for Accessible Communication

Although the initial iteration of the intervention video was highly successful at increasing use of attention getting strategies to elicit infant visual attention, parents continued to use these strategies to redirect infant attention, with an increase in use in some cases. We therefore presented the use of strategies within the context of semantically contingent communication with clearer emphasis on the purpose of the strategies (i.e., to elicit infant attention before communicating). Whilst implementing contingency, the goal for parents was to alert the infant to forthcoming communication and increase access to communication by initially eliciting their attention. To mitigate against the possibility that focusing on implementing visual attention getting strategies may not be the most appropriate for all families (depending on factors such as infant hearing loss levels and use of hearing technology) and subsequently challenging to implement, we made two developments.

Firstly, we added more strategies to increase auditory access to language such as background noise management, which can be challenging for parents [79] and perceived as a less important strategy by parents [31]. We explicitly included close proximity to the child when communicating given that findings from study 1 suggest parents may not be adjusting their position when their child is mobile. We additionally included the recommendation to increase awareness of child position and relocate when the child settles, rather than continuously following the child to avoid exerting control over interaction. The second development was advice for parents on how to find the right balance of visual and auditory strategies when making their communication accessible (i.e., a more tailored approach to better suit child and family specific factors), rather than providing a set of strategies in a ‘one-size-fits-all’ approach. 

### 3.2. Taking a Modular Approach

To address a number of acceptability issues raised in study 1, we changed the intervention to short, themed videos. This allowed for the inclusion of additional strategies (as desired by parents in study 1) without increasing length, which was an issue for some parents in study 1. The current intervention therefore included additional strategies such as expansion and acoustic highlighting [80,81]. We additionally included the strategies of pausing and waiting when communicating to provide increased opportunities for the child to decode utterance meaning and respond, subsequently optimizing the language learning process [82]. Additionally, the modular design provides parents with the option to focus on specific modules rather than all intervention strategies at once. It also allows for professionals to incorporate specific videos into practice when supporting targeted areas of communication as part of a tailored approach, which can be effective [77]. A final development included proving illustrative video clips of families from more varied backgrounds in response to feedback from study 1 where some participants struggled to relate to the parents modeling the strategies.

### 3.3. Materials and Methods

#### 3.3.1. Participants

Participant demographics are detailed in Table 4. Two groups of participants were recruited across the UK: 9 parents of a deaf child and 17 professionals working with families who have a deaf child (e.g., Qualified Teachers of the Deaf [QToDs], Speech and Language Therapists [SLTs]). Professionals were contacted via email inviting them to take part in the present study and to share the invite with the parents they support. A study advertisement was also posted on social media and relevant websites with a view to recruiting both parents and professionals. Parents were eligible to take part if they were over 16 years of age and had a child with any level of permanent hearing loss. Professionals were eligible to take part if they specialized in early years support.

#### 3.3.2. Procedure

All participant procedures were online. Participants who were interested in taking part first followed a link in the email invitation, which directed them to an online participant information sheet and consent form. After participants consented to take part, they were asked to watch the intervention videos and complete the relevant online anonymous acceptability survey detailed below. A link to the intervention videos was available in the email invitation. Ethical approval was granted by the Department of Psychology’s Ethics Sub-Committee at The University of Sheffield.

#### 3.3.3. Intervention

Six videos were created in line with feedback from study 1 to form a series entitled ‘Supporting Communication with Deaf Babies and Toddlers’. Similar to the intervention video in study 1, the revised intervention videos described strategies to support access to communication, scaffold developing communicative skills, and facilitate interaction for families who have a deaf infant or toddler. Unlike study 1, the videos used a modular format where intervention content was broken down into short, content-specific modules (see Table 5 for an overview of the intervention videos), which ranged in length from 3 min, 45 s to 11 min, 52 s. The modular format aimed to provide a wealth of strategies in a more manageable way, whilst making content more flexible in delivery. For example, parents could watch the videos over several sittings, watch those most relevant to the challenges they face, or refresh specific strategies. Similarly, professionals could choose specific videos depending on the needs of the family they are working with. The videos were also designed to be more inclusive of current family communication approach(es) (e.g., aural-oral, SSE, etc.), child hearing status, child hearing technology (e.g., hearing aids, cochlear implants, etc.), and family background. The videos provided a detailed description of strategies through narration over carefully selected illustrative video clips from naturalistic, video-recorded, infant–parent interaction, and supporting animations. Scripts were developed using accessible language by the first author in consultation with remaining authors, other researchers, and professionals who worked closely with families with a deaf child in the early years, including QToDs and specialist SLTs (see Appendix A). 

#### 3.3.4. Measurements

1.Demographic Information

Demographic data collected included: (i) parent characteristics (i.e., relationship to child, hearing status, primary language, communication approach(es), main source of communication support); (ii) child characteristics (i.e., level of hearing loss, age, hearing technology); and (iii) professionals’ characteristics (i.e., role, main approach used to support families, UK country of practice).

2.Intervention Acceptability Survey

Two novel anonymous acceptability surveys were developed, one for parents and one for professionals (20 and 27 items, respectively), which were based on the Theoretical Framework of Acceptability (TFA; [78]). The TFA comprises seven domains that assess the acceptability of interventions including affective attitudes, burden, perceived effectiveness, ethicality, intervention coherence, opportunity-costs, and self-efficacy. As parents and professionals were not asked to implement the strategies or incorporate them into their professional practice (respectively) prior to completion, both surveys provide a prospective assessment of acceptability. Additional items that did not fit within the TFA but were believed to be pertinent to assessing the acceptability of the videos and for future development were included (e.g., ‘*How likely would you be to recommend these videos to other parents*’). Most items were consistent across both surveys; however, additional items were included in the professionals’ survey to assess their perception of the videos’ acceptability for parents. Participants rated the extent to which they agreed with each item using a 7-point Likert scale ranging from 1 (strongly disagree) to 7 (strongly agree).

### 3.4. Results

#### Intervention Acceptability

Parent acceptability of the video intervention is presented in Table 6, while professionals’ perceptions of acceptability are displayed in Table 7.

1.Parent Acceptability

Broadly, parents reported that they felt the video intervention was acceptable (see Table 6), with positive median ratings for each sub-scale of the acceptability survey. Parents reported that they found the intervention videos to be engaging (median = 6), enjoyable (median = 6), and not too burdensome in terms of length, level of information, and potential difficulty implementing the strategies (all medians = 2). Parents also reported that the intervention videos were compatible with the way they would like to support their child (median = 6), that they were understandable and coherent (all medians = 6), and would not be too onerous to use in terms of time and resources (medians = 2). Finally, parents reported that they felt the intervention videos would be effective (median = 6), that they felt confident in their own ability to use the intervention strategies (medians ≥ 5), that they think using a video is a good way to provide information (median = 6), and that they would recommend the intervention to other parents (median = 7).

2.Professionals Acceptability

Professional acceptability (Table 7) was also high across all subscales. Professionals reported that they felt the videos were engaging and enjoyable (medians = 7), and that parents would think the same (medians = 7). Professionals also reported that the intervention videos would not be burdensome or difficult to implement (all medians ≤ 2), that they fit well with how they think families should be supported (median = 7), and that the intervention videos were understandable and coherent (medians = 7). Professionals reported that the intervention videos would be worth the time and resources to administer (median = 6), would likely be effective for parents and professionals (medians ≥ 6), that using videos is a good way to provide information (median = 7), and that they would likely use the videos in their normal practice (median= 7). However, although professionals reported feeling confident in their ability to use the videos with families (median = 7), they reported less confidence that parents would have the confidence to implement the intervention strategies (median = 5).

### 3.5. Discussion

The new iteration of the intervention was considered acceptable by both parents and professionals. Parents felt the videos were a good way to provide information about supporting communication development, and that they would recommend them to other parents. Further, parents felt the videos were not difficult to understand and if they used the strategies, they would not feel uncomfortable or find them difficult to implement. However, min-max scores suggest some parents may find the videos too difficult to understand and the strategies too uncomfortable to use, raising potential barriers to implementation. Another potential barrier raised was parent confidence in using the videos to support their child, as this item had the lowest median acceptability score for both parents and professionals. Further investigation into potential barriers is an important consideration for future intervention development to increase the likelihood of intervention effectiveness [83]. One area of acceptability showing some improvement from study 1 is video length. Although considered largely acceptable in both studies, there was less variability in the present study with more parents disagreeing the videos were too long, suggesting the new modular format is more acceptable. 

Professionals’ acceptability scores parallel parents’, with agreement across items exploring professionals’ expectations of parent acceptability. These findings have positive implications for a future standardized intervention that is acceptable for use in routine practice. Indeed, by reaching consistent acceptability across groups there is an increased likelihood of parental engagement and intervention efficacy, as well as implementation into practice [84]. It is important to note; however, that some acceptability scores for professionals suggest there may be lower acceptability for a minority. Given the importance of professionals’ acceptability for the long-term success of the intervention, further work is needed to understand potential acceptability issues and develop the videos accordingly [84]. 

## 4. General Discussion

The present research demonstrated feasibility and acceptability of an iteratively developed video-based intervention designed to promote specific communication strategies to support infant–parent interaction with a deaf infant. Attention now turns to testing the efficacy and acceptability of the intervention in a more methodologically rigorous, randomized controlled pilot trial with a larger sample size. A randomized pilot trial will allow us to test the efficacy of the intervention relative to a control group, which in turn gives us more accurate estimates of the intervention effects [85] on outcomes that are important to both parents and professionals. Furthermore, conducting a randomized pilot trial will inform key methodological parameters that are important when planning future larger, confirmatory trials (e.g., effect sizes, possible attrition rates, practical challenges, etc.). It is also important to note that intervention development is a continual, iterative process [84]. Therefore, it is likely that future pilot work will result in further intervention changes and refinement based on acceptability feedback and stakeholder involvement. Finally, it is vital that intervention development attempts consider practical implementation of the intervention early in the process. Failure to do so will likely lead to an intervention that is not appropriate for use in practice, and therefore will suffer from poor engagement and limited effectiveness [86]. The present research investigated some factors that might impact implementation later down the line; however, future research should consider implementation using a more comprehensive and theory-driven approach such as the Theoretical Domains Framework [87]. Doing so at this early stage will illuminate a range of barriers that might limit implementation (and therefore effectiveness) and allow these barriers to be addressed pre-emptively.

### Strengths and Weaknesses

The present research has a number of strengths. Firstly, we have taken a first step in the intervention development life cycle by conducting a comprehensive and iterative study of acceptability and feasibility that lays the groundwork for further intervention development. Secondly, a range of parent and professional stakeholders provided input during the design stage, a key factor in successful implementation and future engagement. Third, we have taken a fine-grained approach to the measurement of outcomes at the behavioral level. Doing so allows greater understanding of how the intervention impacted specific behaviors (i.e., the mechanisms of outcome change), rather than measuring possible outcomes of those behaviors (e.g., child language). Finally, we have made the intervention videos freely available to parents and professionals via NDCS, which is the leading UK charity for childhood deafness [88]. However, there are limitations to consider. Future research should recruit a more representative sample than the present studies to ensure findings are applicable to a more diverse range of families. For example, including more fathers, who may uniquely scaffold their children’s language development [89,90], and might have lower parenting self-efficacy than mothers [91]. Additionally, a more representative sample in terms of degree of hearing loss would allow for exploration into any potential effects of this variable on feasibility and acceptability. Finally, including families of infants using other hearing technologies (e.g., cochlear implants) and infants with additional needs would also be beneficial to determine whether adaptations are required to reach feasibility and acceptability within these groups.

## 5. Conclusions

In conclusion, the present research aimed to investigate the feasibility and acceptability of a video-based intervention designed to increase use of communicative strategies in parents of deaf infants. Overall, the intervention was acceptable to parents and professionals, although further refinements could be addressed in future interventions. The intervention promoted many parent communicative behaviors and represents a positive early step towards a standardized intervention to support communication that could be used in routine practice. 

## Figures and Tables

**Figure 1 jcm-11-05272-f001:**
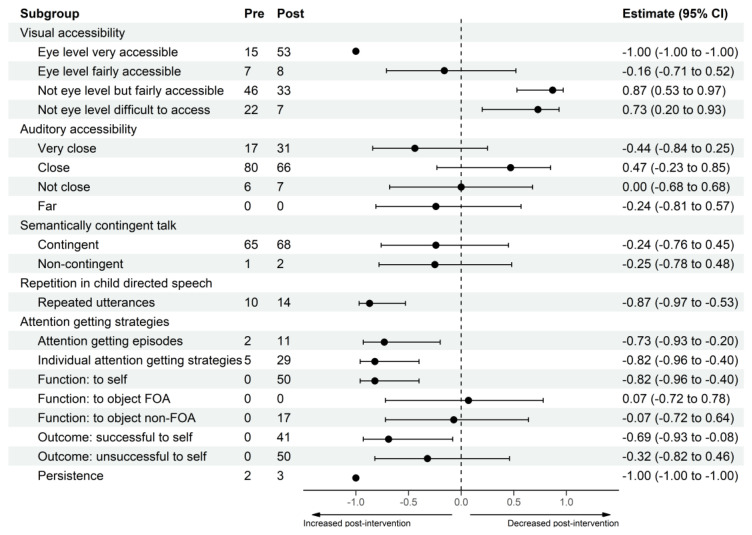
Forest Plot Showing Median Proportions Pre- and Post-intervention and Rank-Biserial Effect Size Differences from Pre- to Post-intervention for all Communication Strategies Measured. Note. FOA = focus of attention.

**Figure 2 jcm-11-05272-f002:**
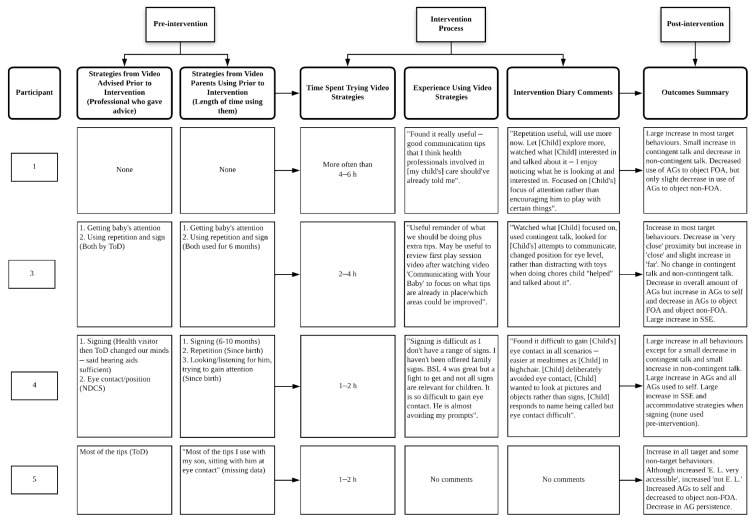
A Process Evaluation Flow Diagram Describing per Participant Experiences of the Intervention. Note. AG = attention getting strategies. FOA = focus of attention. QToD = Qualified Teacher of the Deaf. SSE = Sign Supported English. NDCS = National Deaf Children’s Society. BSL = British Sign Language. E. L. = eye level. SLT = Speech and Language Therapist.

**Table 1 jcm-11-05272-t001:** Participant Characteristics and Background Information of Deaf Infant–Hearing Parent Dyads.

Participant	Relationship	Infant Age ^a^ 1st Visit	Hearing Loss Left Ear	Hearing Loss Right Ear	Age HAs Received	Home Communication	Parent Qualifications	Primary Caregiver	IMD Decile
1	Mother	12 m 21 d	Moderate-Severe	Moderate	5 weeks	Spoken English	Mother: 6 Father: --	Mother	6
3	Mother	13 m 7 d	Moderate-Severe	Profound	15 weeks	Mostly SSE	Mother: 5 Father: 2	Mother	5
4	Mother	19 m 9 d	Moderate	Moderate	6 weeks (also using FM system)	Spoken English, some SSE	Mother: 6 Father: --	Mother	6
5	Mother	18 m 17 d	Severe	Severe	12 weeks	Spoken English	Mother: 2 Father: 3	Mother	4
6a	Mother	12 m 25 d	Profound	Severe	12 weeks	Spoken English, some SSE	Mother: 6	Mother	6
6b	Father	As above	As above	As above	As above	As above	Father: 6	As above	As above
7	Mother	18 m 12 d	Moderate	Moderate	15 weeks	Spoken English	Mother: 6 Father: 6	Mother	2
8	Mother	18 m 20 d	Moderate	Severe	8 weeks	Spoken English	Mother: 8 Father: 8	Mother	7
9	Mother	12 m 30 d	Severe-Profound	Moderate-Severe	9 weeks	Spoken English and SSE	Mother: 6 Father: 6	Mother	7

Note. IMD Decile = Indices of Multiple Deprivation Decile (where areas considered to be within the most deprived 10% of England and Scotland = 1, and areas considered to be within the least deprived 10% of England and Scotland = 10). HA = hearing aids. SSE = Sign Supported English. FM = frequency modulation. Parent qualifications were determined by UK Government guidelines where 1 = GCSEs (grades D-G), NVQ level 1; 2 = GCSEs (grades A*–C), NVQ Level 2; 3 = A Levels, NVQ Level 3; 4 = NVQ Level 4; 5 = NVQ Level 5; 6 = Bachelor’s Degree with Honors; 7 = Postgraduate Degree/Certificate/Diploma; 8 = Doctorate. -- = missing/incomplete data. ^a^ Age is in months and days.

**Table 2 jcm-11-05272-t002:** Overview of Pre-Post Changes in Communication Strategies.

							95% CI for Rank-Biserial	
Communication Strategies	Median Pre	Median Post	W	Z	*p*	Rank-Biserial	Lower	Upper
Visual accessibility of language								
Eye level very accessible	15	53	0.00	−2.67	0.009	−1.00	−1.00	−1.00
Eye level fairly accessible	7	8	19.00	−0.42	0.721	−0.16	−0.71	0.52
Not eye level but fairly accessible	46	33	42.00	2.31	0.020	0.87	0.53	0.97
Not eye level difficult to access	22	7	39.00	1.96	0.058	0.73	0.20	0.93
**Auditory accessibility of language**								
Very close	17	31	12.50	−1.19	0.260	−0.44	−0.84	0.25
Close	80	66	33.00	1.24	0.235	0.47	−0.23	0.85
Not close	6	7	14.00	0.00	1.000	0.00	−0.68	0.68
Far	0	0	8.00	−0.52	0.674	−0.24	−0.81	0.57
**Semantically contingent talk**								
Contingent	65	68	17.00	−0.65	0.553	−0.24	−0.76	0.45
Non-contingent	1	2	13.50	−0.63	0.570	−0.25	−0.78	0.48
**Repetition in child directed speech**								
Repeated utterances	10	14	3.00	−2.31	0.024	−0.87	−0.97	−0.53
**Attention getting strategies**								
Attention getting episodes	2	11	6.00	−1.96	0.057	−0.73	−0.93	−0.20
Individual attention getting strategies	5	29	4.00	−2.19	0.027	−0.82	−0.96	−0.40
Function: to self	0	50	4.00	−2.19	0.033	−0.82	−0.96	−0.40
Function: to object FOA	0	0	8.00	0.14	1.00	0.07	−0.72	0.78
Function: to object non-FOA	0	17	13.00	−0.17	0.93	−0.07	−0.72	0.64
Outcome: successful to self	0	41	5.50	−1.75	0.092	−0.69	−0.93	−0.08
Outcome: unsuccessful to self	0	50	9.50	−0.76	0.498	−0.32	−-0.82	0.46
Persistence	2	3	0.00	−2.37	0.022	−1.00	−1.00	−1.00

Note. FOA = focus of attention.

**Table 3 jcm-11-05272-t003:** Acceptability of Intervention Content and Mode of Delivery.

Item	Median	IQR	Min	Max
**Section 1: Acceptability of Communication Strategies** ^a^				
I enjoyed trying the tips from the video	8.5	2.25	4	10
I did not find it difficult to use the tips in my daily routine	9	1.00	1	10
I felt comfortable trying the tips from the video	10	0.25	2	10
The tips helped me communicate with my baby	7.5	2.25	7	10
I will continue to use the tips from the video going forward	10	0.00	10	10
**Section 2: Acceptability of Video Delivery** ^b^				
The video length was too short/long ^c^	5.5	2.50	5	8
The amount of information was… too little/too much ^d^	5	0.75	2	5
The video was… difficult/easy to understand ^e^	10	0.00	9	10
The video was… not professional/very professional ^f^	8.5	1.75	8	10
Using a video is a good way to provide advice to parents ^g^	10	1.50	7	10
I would recommend this video to other parents ^g^	7.5	2.50	5	10

Note. ^a^ In Section 1, higher scores = high acceptability. ^b^ Section 2 used visual analogue scales, therefore interpretation varies, see notes for details. ^c^ Lower scores indicate too short and higher indicates too long. ^d^ Lower scores indicate too little information and higher indicates too much information. ^e^ Lower scores indicate difficult to understand and higher indicates easy to understand. ^f^ Lower scores indicate not professional, higher scores indicate very professional. ^g^ Higher scores indicate higher agreement with the item. IQR = Interquartile range, Min = minimum score, Max = Maximum score.

**Table 4 jcm-11-05272-t004:** Participant Characteristics and Background Information.

Parent Characteristics	*N*	Child Characteristics	** *N* **
Relationship to deaf child		Level of hearing loss	
Mother	8	Mild-moderate	1
Father	1	Moderate	1
Hearing loss		Moderate-severe	3
Yes	1 (mild)	Profound	4
No	8	Age	
Primary language		7–12 months	2
English	6	13–18 months	2
British Sign Language	2	19–35 months	3
Spanish	1	5–11 years	1
Communication Approach(es)		12+ years	1
Auditory-oral	3	Hearing technologies	
Sign language	2	Hearing aid(s)	7
Sign/speech bilingualism	4	Bone-anchored hearing aid(s)	1
Sign supported English	2	FM System	2
Makaton/PECS	1	None	1
Main source of communication support			
QToD	8		
Media	2		
Local support groups	3		
Deaf charity resources	3		
**Professionals’ Characteristics**	** *N* **		
Professional role			
QToD	6		
SLT	9		
AVT Therapist	1		
Paediatric Nurse	1		
Main approach used to support families			
Aural-oral	1		
Mix of approaches (case-by-case)	14		
Sign/speech bilingualism	1		
AVT	1		
UK country of practice			
England	14		
Wales	3		

Note. PECS = Picture Exchange Communication System. QToD = Qualified Teacher of the Deaf. FM = frequency modulation. SLT = Speech and Language Therapist. AVT = Auditory Verbal Therapy.

**Table 5 jcm-11-05272-t005:** Overview of Intervention Videos.

Video Title	Video Description
Introduction video: An Introduction to Supporting Early Communication Development	Introduces the video series by providing an overview of the videos and contextualizing the resource, emphasizing that content is designed to fit within different communication approach(es).
Video 1: How do Babies Learn to Communicate?	Provides a brief overview of general communicative development during the first two years of life, introducing families to important topics such as ‘turn-taking’ and ‘joint attention’.
Video 2: Supporting the Communication Development of Deaf Babies and Toddlers	Presents specific considerations for the communicative development of 0–2-year-olds with any level of hearing loss, namely the increased likelihood of missed opportunities for language learning due to a reduced access to spoken language.
Video 3: Tuning-in and Responding to Your Baby’s Communication	Focuses on encouraging caregivers to contingently comment on child’s focus of attention and attempts to communicate, whilst acknowledging that even though parents will likely already be implementing this strategy, actively doing this will increase learning opportunities. Emphasis is placed on eliciting infant attention prior to communicating and increasing access to language.
Video 4: Supporting Access to Language	Describes auditory and visual strategies to support infant access to language, such as managing background noise and getting down to eye level, as well as how to balance using these strategies.
Video 5: Thinking About How We Communicate	Provides specific strategies that may be particularly beneficial when communicating with a deaf infant/toddler including acoustic highlighting, and pausing and waiting.

**Table 6 jcm-11-05272-t006:** Parent Acceptability.

Item	Median	IQR	Min	Max
Affective attitude				
I found the videos engaging	6	1.00	4	7
I thought the videos were enjoyable to watch	6	1.00	4	7
**Burden**				
I found the length of the videos to be too short	2	0.50	1	3
I found the length of the videos to be too long	2	1.50	1	4
I think the videos present too much information to take in	2	1.00	1	4
I think the videos were too difficult to understand	2	1.00	1	7
I think the advice in the videos will be too difficult to carry out	2	2.00	1	4
I think I will feel uncomfortable trying the tips in the videos	2	1.00	1	7
**Ethicality**				
These videos fit well with the way I would like to support my child	6	1.00	4	7
**Intervention coherence**				
I understood the purpose of the videos	6	1.00	5	7
I felt the videos were well structured so that, as a whole, they can help me support the language and communication development of my deaf child	6	0.50	5	7
I felt the content of the videos was understandable	6	1.00	5	7
**Opportunity costs**				
I feel like I would have to give up a significant amount of resources to use and apply the video tips that are relevant to my family	2	0.50	1	3
I feel that using the video tips would go against my preferences for supporting the communication development of my deaf child	2	1.00	1	3
**Effectiveness**				
I think the videos are likely to help me to support my deaf child’s communication development	6	1.00	4	7
**Self-efficacy**				
I feel able to use and apply the video tips that are relevant to my family	6	1.50	5	7
I feel confident using the videos to support my child	5	1.50	4	7
**Other**				
Using videos is a good way to provide information and advice to parents	6	1.00	6	7
How likely would you be to recommend these videos to other parents?	7	1.50	5	7
How likely are you to follow the tips in the videos?	6	1.50	5	7

Note. All items scored from 0 (strongly disagree with the item) through to 7 (Strongly agree with the item). IQR = Interquartile range.

**Table 7 jcm-11-05272-t007:** Professionals Acceptability.

Item	Median	IQR	Min	Max
Affective attitude				
I found the videos engaging	7	1.00	5	7
I thought the videos were enjoyable to watch	7	1.00	5	7
I think that families will find the videos engaging	7	1.00	5	7
I think that families will find the videos enjoyable	7	1.00	4	7
**Burden**				
I found the length of the videos to be too short	1	1.00	1	6
I found the length of the videos to be too long	2	2.00	1	7
I think the videos present too much information to take in	2	2.00	1	6
I think parents will find the videos too difficult to understand	1	1.00	1	5
I think the advice in the videos will be too difficult for families to implement	2	1.00	1	4
I think parents will feel uncomfortable trying the tips in the videos	2	1.00	1	7
**Ethicality**				
The information and advice in these videos fits well with the way I think we should be supporting families	7	1.00	1	7
The information and strategies in these videos are compatible with the approach/mix of approaches I use to support the communication development of deaf infants	7	0.00	6	7
**Intervention coherence**				
I understood the purpose of the videos	7	0.00	7	7
I felt the videos were well structured so that as a whole, they reach their aim of helping parents to support the language and communication development of their deaf infant and/or toddler	7	0.00	4	7
I felt the content of the videos was understandable for English-speaking families where language would not be a barrier	7	1.00	5	7
**Opportunity costs**				
I think that the potential costs to me as a practitioner in terms of time and stress to use these videos while supporting families is outweighed by the benefits I would gain	6	3.00	1	7
I think that the potential costs to families in terms of time and stress to engage with these videos is outweighed by the benefits they could gain	6	2.00	1	7
**Effectiveness**				
I think the videos are likely to achieve their purpose	6	1.00	4	7
Videos are a useful tool to use in practice to provide information and strategies for families with a deaf child/children	7	1.00	4	7
I think the videos will be a useful tool for practitioners who support families	7	1.00	6	7
**Self-efficacy**				
I feel confident in helping families to understand the content of these videos	7	0.00	6	7
I feel confident in helping families to adapt the advice in these videos to fit their personal circumstances	7	0.00	6	7
I feel confident in helping families to feel motivated and able to engage with these videos to support their deaf child/children	7	1.00	5	7
I think that families will have the confidence to put the advice in the videos into action	5	1.00	4	7
**Other**				
Some families will need support to understand and implement the information and strategies in the videos in their day-to-day lives	7	0.00	5	7
Using videos is a good way to provide information and advice to parents	7	0.00	6	7
How likely are you to use these videos in your practice with families who have a young deaf child?	7	1.00	2	7

Note. All items scored from 0 (strongly disagree with the item) through to 7 (Strongly agree with the item). IQR = Interquartile range.

## Data Availability

The data that support the findings of this study are available from the corresponding author, (C.K.) upon reasonable request.

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
