# Peer review of "Assessing a Video-Based Intervention to Promote Parent Communication Strategies with a Deaf Infant: A Feasibility and Acceptability Study"

_jcm, 2022, doi:10.3390/jcm11185272_

Round 1
Reviewer 1 Report
I find the manuscript interesting for readers and I really enjoyed reading it. The manuscript is of sufficient quality, scientifically sound and methodologically flawless. Here are some comments for further improvements:
- Please, correct some typos, e.g. ref 1- first line, parentheses of references
- Please, discuss validity and reliability of the measuring instrument.
- Please, substantiate the unbalanced sample in terms of hearing loss.
- Implications for practice are clear. However, I would suggest elaborating theoretical implications of the study.
Author Response
Dear Reviewer 1,
Thank you for your positive comments and suggestions for further improvements to our manuscript. We have addressed concerns and updated the manuscript in line with suggestions. Please find a detailed response below.
Please, correct some typos, e.g. ref 1- first line, parentheses of references
Thank you for raising this issue. We have read through the manuscript for typos and errors, which have now been corrected, including the typo raised in reference 1.
Please, discuss validity and reliability of the measuring instrument.
We felt that psychometric validation would not be appropriate in the present study as the coding scheme (which the measures are based on) was designed to detect observable events (e.g. repetition of speech) rather than latent constructs. In relation to reliability, we report reliabilities between independent coders on the bottom of page 13 of the updated manuscript. As all reliabilities were substantial/excellent), we believe the measuring instrument to be reliable. We have added: Agreement between raters suggests measures are reliable to the end of the ‘Reliability of Coding’ section (highlighted top of page 14).
Please, substantiate the unbalanced sample in terms of hearing loss.
Thank you for raising this. We have now discussed the unbalanced sample in terms of hearing loss in the ‘Strengths and weaknesses’ section of the General Discussion by including the text: Additionally, a more representative sample in terms of degree of hearing loss would allow for exploration into any potential effects of this variable on feasibility and acceptability (highlighted bottom page 33).
Implications for practice are clear. However, I would suggest elaborating theoretical implications of the study.
Thank you for this suggestion. We chose not to discuss theoretical implications as the research questions are practical rather than theoretical in that they aim to investigate an applied clinical problem. We aimed to provide theoretical justification for our approach and believe we would be able to say more in terms of theory if the intervention were tested with a larger sample, and child language outcomes were assessed. We therefore feel there aren’t any specific theoretical conclusions that can be appropriately derived from the research but we are open to suggestions.

Reviewer 2 Report
This is a prospective cohort study aimed at assessing a video-based intervention to promote parent communication strategies with a deaf infant, designed as a feasibility and acceptability study.
The abstract is adequate, and has listed rationale and setting details, alongside the most important findings.
The objectives of the study are presented clearly and the introduction section communicates the need for defining tools in encouraging parent-child interactions is the setting of a deaf infant and hearing parents. However, the introduction, although fascinating, is much too long and unstructured. The majority of the section titled Interaction at Risk: Deaf Infant-Hearing Parent Dyads, should be transferred to the Discussion section, if we are to maintan the IMRAD organization. The section defining Current Practice and Problems should be the last paragraph in the introduction, defining the inervention and its aims. The rest of the overly long introduction section and subsections should be abbreviated, as they read repetitive, unfocused and vague. The point of this paper is not to inform the reader on every aspect of deaf newborn rehabilitation, but rather to focus on a specific intervention.
Then, a Materials and Methods section has to clearly defined inclusion and exclusion criteria for both studies. It has listed adequate statistical tests and questionnaires.
The Results section communicated the findings well, but should limit the number of tables and figures to 6 total. The division of two studies with their separate MM, Results and Discussion sections is not acceptable.
The discussion an a honest limitations and shortcomings paragraph and the results are fascinating. I would congratulate the authors on a detailed and interesting discussion.
Author Response
Dear Reviewer 2,
Thank you for your feedback, comments, and suggestions to improve our manuscript. We have addressed concerns and updated the manuscript in line with suggestions. Please find a detailed response below.
The objectives of the study are presented clearly and the introduction section communicates the need for defining tools in encouraging parent-child interactions is the setting of a deaf infant and hearing parents. However, the introduction, although fascinating, is much too long and unstructured.
Thank you for raising this issue. We have edited the manuscript so that the introduction is now considerably shorter, more structured, and concise. The introduction now briefly introduces the importance of infant-parent interaction and the challenges faced by parents of deaf infants when scaffolding development and making communication accessible, followed by a more concise section on current intervention practice and problems, ending with the aims of the research. We have additionally made minor edits throughout manuscript to ensure the whole manuscript is concise.
The majority of the section titled Interaction at Risk: Deaf Infant-Hearing Parent Dyads, should be transferred to the Discussion section, if we are to maintain the IMRAD organization.
In order to reduce the length of the introduction and to be more structured and concise, we removed this section from the manuscript. However, we summarised the key points in terms of the challenges parents face scaffolding development and making communication accessible, and incorporated it into the introduction (page 4) as opposed to the discussion, to highlight the issues raised in existing research that the present intervention aims to target.
The section defining Current Practice and Problems should be the last paragraph in the introduction, defining the intervention and its aims.
We have edited the manuscript so that the ‘Current Intervention Practice and Problems’ section is now the last paragraph of the introduction (page 5) prior only to 'The present studies' section, which outlines the research aims.
The rest of the overly long introduction section and subsections should be abbreviated, as they read repetitive, unfocused and vague. The point of this paper is not to inform the reader on every aspect of deaf newborn rehabilitation, but rather to focus on a specific intervention.
Thank you for this suggestion. We have removed the remaining section of the introduction ‘Initial development of an intervention to promote the use of specific communication strategies: Rationale’ (and subsections) so that the newly edited introduction remains more focussed and less of a narrative review of the literature. The content of this section has been considerably reduced and moved to the Materials and Methods section under the subsection ‘Intervention’ in order to provide a brief summary of the included intervention strategies. The summarized text can be found on pages 7-8 under the subheadings: Strategies for scaffolding infant skill development, Strategies for making communication accessible: Deaf parents as role models, Additional potentially beneficial strategies.
The Results section communicated the findings well, but should limit the number of tables and figures to 6 total.
Thank you for this positive feedback. We agree that 6 tables and figures would be more appropriate for a single study. However, as the present research is two distinct studies (please see reasoning below), we felt 9 tables and figures is appropriate, but we are happy to defer this decision to the Editorial team.
The division of two studies with their separate MM, Results and Discussion sections is not acceptable.
Although the present study reports on the iterative development of a parent intervention, we created two individual interventions that were tested with two different samples of participants, using different methods. In line with similar studies (e.g., Thoen & Robitschek, 2013)* we therefore presented the research as two distinct studies, which we believe is necessary to ensure clear reporting of the present research. However, if Reviewer 2 has any specific details outlining how the manuscript structure is not acceptable, we would be happy to consider this in a revised version.
The discussion an a honest limitations and shortcomings paragraph and the results are fascinating. I would congratulate the authors on a detailed and interesting discussion.
We thank you for your positive feedback.
*Thoen, M. A., & Robitschek, C. (2013). Intentional Growth Training: Developing an Intervention to Increase Personal Growth Initiative. Applied Psychology: Health and Well‐Being, 5(2), 149-170.

Round 2
Reviewer 2 Report
The paper has benefited from the revision in terms of clarity, argument flow and organization. I would support the submission for publication.